# Scale dependence in hydrodynamic regime for jumping on water

Minseok Gwon [1], Dongjin Kim[1], Baekgyeom Kim [1], Seungyong Han [1] ✉, Daeshik Kang [1] ✉ & Je-Sung Koh [1] ✉

Momentum transfer from the water surface is strongly related to the dynamical scale and morphology of jumping animals. Here, we investigate the scale-dependent momentum transfer of various jumping organisms and engineered systems at an air-water interface. A simplified analytical model for calculating the maximum momentum transfer identifies an intermediate dynamical scale region highly disadvantageous for jumping on water. The Weber number of the systems should be designed far from 1 to achieve high jumping performance on water. We design a relatively large water-jumping robot in the drag-dominant scale range, having a high Weber number, for maximum jumping height and distance. The jumping robot, around 10 times larger than water striders, has a take-off speed of 3.6 m/s facilitated by drag-based propulsion, which is the highest value reported thus far. The scale-dependent hydrodynamics of water jumpers provides a useful framework for understanding nature and robotic system interacting with the water surface.

Semi-aquatic animals, such as water striders and fisher spiders, exhibit a unique locomotion mechanism involving jumping on water distinguished from jumping on ground[1–4]. These animals overcome large obstacles or escape from predators by leaping out of the water surface. In the general case of jumping on rigid ground, a high actuation power density, corresponding to high speed and force, and large-scale driving appendages for longer propulsion times lead to high and long jumps from the ground[5–8], but this principle is not applicable to jumping from the water surface. In the mechanics of jumping on water, the water striders and water strider-inspired robots experienced meniscus breaking on the water surface when the leg speed increased, resulting in decreased jumping performance[1,9]. In contrast, Pygmy mole crickets (Body length ($L$) = 6 mm, Leg speed ($U$) = 6.12 m/s, take-off velocity ($V$) = 1.3 m/s), which are smaller than the water striders ($L$ = 28 mm, $U$ = 1.05 m/s, $V$ = 1.5 m/s), jump faster from the water surface utilizing their faster leg speed and paddles attached to the end of the legs[10]. There have been studies to elucidate the jumping mechanism on the water for specific species[1,3,11], but few studies have been performed to find an optimal jumping performance related to scale and hydrodynamics considering overall jumping animals and robots.

Several robots inspired by nature organisms that locomote on the water surface have been developed to understand dynamic interaction at the air-water surface or to expand the working range of insect-scale robots[12–16]. Their size and legs are designed to utilize the dominant hydrodynamic force experienced by their natural counterparts. Small-scale robots that rely on surface tension for floating on water inspired by water striders have demonstrated the ability to jump vertically on the water by maximizing momentum against the surface tension force[1]. Comparatively large-scale robots inspired by basilisk lizards running on water generate propulsive force associated with inertial drag[17–19]. However, overcoming obstacles is still challenging due to low performance or the absence of a directional jump. By comparison with natural organisms on the water surface, design principles can be found that maximize the momentum for water jumping.

We investigate how the jumping performance of animals and robots on the water surface depends on the morphological and dynamic scales involved. We derive a simplified model that describes a jumping mechanism on water, which delineates the scaling of principle components. The analytical results from this jumping model show a non-linear relationship between system size and take-off velocity and reveal a dynamical scale range that does not facilitate large momentum

[1]Department of Mechanical Engineering, Ajou University, 206 Worldcup-ro, Yeongtong-gu, Suwon-si, Gyeonggi-do 16499, Republic of Korea.
✉e-mail: sy84han@ajou.ac.kr; dskang@ajou.ac.kr; jskoh@ajou.ac.kr

from the water surface. Based on the simplified jumping model, we design vertical and directional jumping robots with a superior jumping performance by maximizing the dominant hydrodynamic force. The 3 g jumping robot demonstrates a maximum leap height of 545 mm from the water surface, and most of the propulsion is generated by the drag force. The take-off velocity of the robot is 3.6 m/s, which is the highest jumping performance from the water surface ever reported for animals and robots that jump on water (Fig. 1). As a practical application of jumping on water, we demonstrate that the robot can jump over an obstacle 220 mm high on the water surface by directional jumping. (Supplementary Movie 1). This robotic platform also provides experimental insights and motivation for studying morphological evolution in nature.

## Results
### Scaling of jumping on water
In nature, animals and robots that locomote on the water surface experience different types of dominant hydrodynamic forces. Here, we show the relation between hydrodynamic forces and their scales and present a model of jumping performance depending on the dominant hydrodynamic force. Previous studies classified semi-aquatic animals that move or jump on the water surface according to the magnitude of dimensionless numbers: The Baudoin (Ba) number, Weber (We) number, and Bond (Bo) number[20,21]. These dimensionless numbers identify the relative magnitudes of forces exhibited by animals and robots moving on the water surface. The Weber number (We $= \rho U^2 L_c/\sigma$, where $\rho$ is the density of the fluid, $U$ is the velocity of the legs, and $L_c$ is the characteristic length) is the ratio of the inertial drag to the surface tension force, the Bond number (Bo $= \rho g L_c^2/\sigma$, where $g$ is the gravitational acceleration) is the ratio of the buoyancy to the surface tension force, and the Baudoin (Ba $= Mg/\sigma P$, where $M$ is the mass, $g$ is the gravitational acceleration, $\sigma$ is the surface tension of water, and $P$ is the contact perimeter) number represents their morphological scale associated with their size of water-jumping animals and robots[15]. Figure 1a, b shows the dependence of these dimensionless numbers for animals[1,3,4,10,11,13,15,20,22–24] and robots[9,13,15,25] jumping on

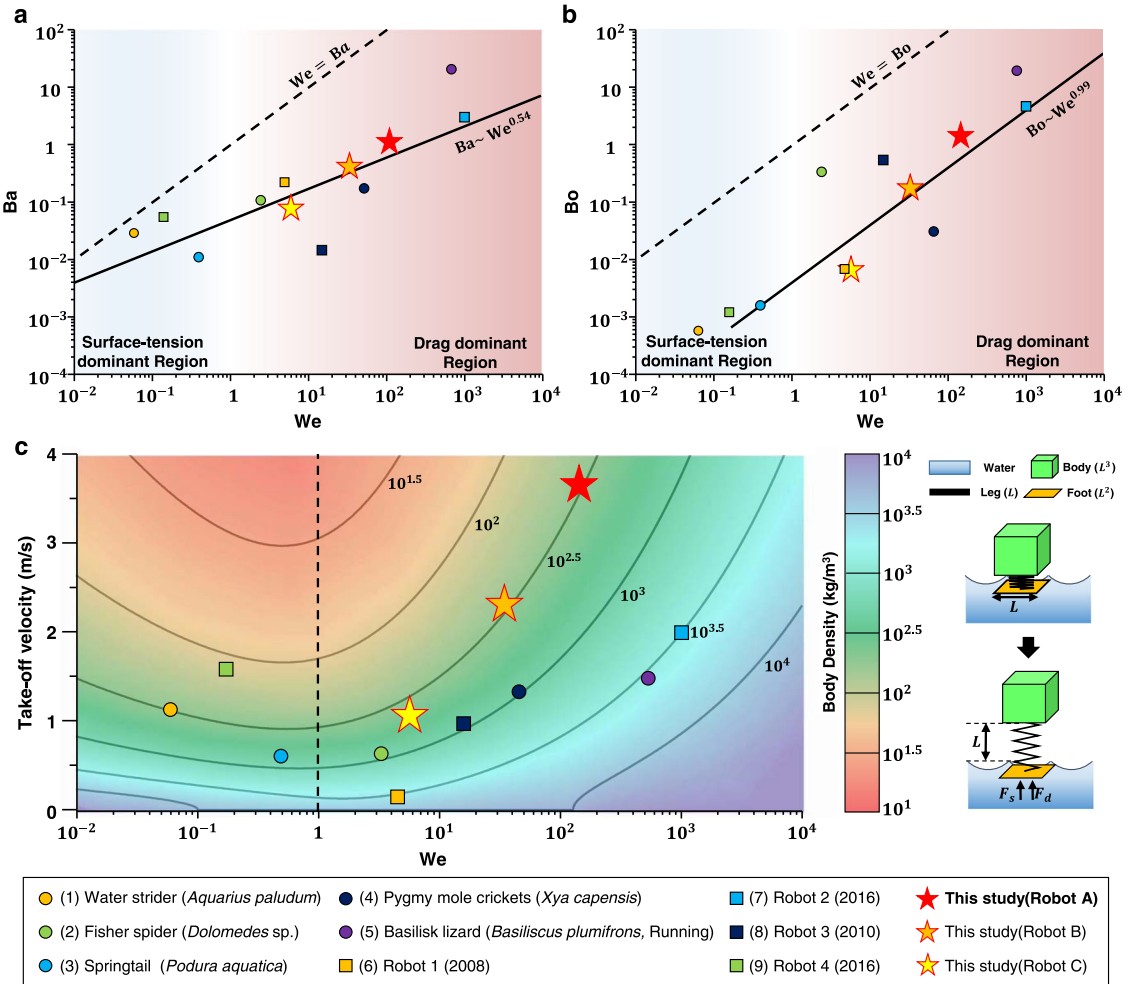

**Fig. 1 | The trend of the Baudoin (Ba), Bond (Bo), and take-off velocity of jumping animals and robots depending on Weber (We) number. a, b** Baudoin (Ba), Weber (We), and Bond (Bo) number for insects and robots jumping on water. Bold lines indicate the dependence of the best-fit line (Ba = 0.0496 We^0.54, $r^2 = 0.597$, Bo = 0.0038 We^0.99, $r^2 = 0.7221$) and dashed line indicate the Ba=We and Bo=We, respectively. The blue region indicates that the jumping locomotion of systems is dominated by surface tension rather than drag (We≪1), and the systems locomotion in the red region is dominated by drag rather than surface tension. **c** The take-off velocity of the insects and robots depends on the Weber number.

Color contour indicates the body density (kg/m³) on the Weber number simulated by a simple model consisting of three components: body (L³), foot (L²), and leg (L). Black lines indicate that the take-off velocity of the model with various body densities varies with the Weber number. Data for characteristic leg speed U, area A, width w, and the take-off velocity collected from (1) Yang et al. (2016), Hu et al. (2010) and Hu et al. (2003) (2) Suter et al. (2000) and Shin et al. (2008) (3) Sudo et al. (2015), Hu et al. (2010) and Kim et al. (2017), (4) Burrows et al. (2012), (5) Bush et al. (2006), Hsieh et al. (2003) and Glasheen et al. (1996) (6) Shin et al. (2008), (7) Yang et al. (2016), (8) Hu et al. (2010), (9) Koh et al. (2015).

water. There is a positive correlation between the Baudoin number and Weber number of various organisms and robots jumping on water. The water surface-based propulsion of small animals and robots with a low Baudoin number (Ba « 1), such as water striders and springtails, is dominated by surface tension forces[4,22] (We < 1). In contrast, comparatively large animals[11,26] and robots[19] (Ba > 1) achieve momentum mainly by the drag force, which is much greater than the surface tension force when these systems jump because of the larger length, size, and speed scales involved. The propulsive force of animals and robots during take-off is generated by surface tension and drag rather than buoyancy, as shown in Fig. 1b (We > Bo), and the viscosity force can be negligible compared with the drag force since most animals and robots locomote on the water surface are characterized by the high Reynolds number[20]. Therefore, the scale comparisons of dimensionless numbers indicate that the dominant hydrodynamic force of jumpers can be classified as drag or surface tension, and the Weber number represents morphological and dynamical scales associated with the size and speed of the legs which determine the dominant hydrodynamic forces.

We derive the analytical model representing jumping performance on the water at various system scales to better understand the momentum transfer through the dominant hydrodynamic force. The take-off velocity of the model visualized with color contours is calculated by a simplified quasi-static model consisting of three components, as shown in Fig. 1c: leg ($L$), foot ($L^2$), and body ($L^3$), representing a length, area, and volume, respectively. The hydrodynamic reaction force is derived considering the surface tension and drag force, which are the dominant forces (see Methods section for details). The plot in Fig. 1c shows the take-off velocity of semi-aquatic animals and robots and model results computed by the simplified model with various Weber numbers. We find that animals and robots in the intermediate-scale region (We ~1), where there is no dominant force, show a worse jumping performance than animals and robots in the surface tension-dominant region (We « 1) or drag-dominant region (We » 1). Fisher spiders and water strider-inspired robots in the intermediate-scale region exhibit worse jumping performance than the other animals and robots we investigate (Fig. 1c and Supplementary Table 1). The take-off velocity of the system is limited by the maximum hydrodynamic reaction force acting on the foot, even if the muscles can exert sufficient force. In this range, due to the low hydrodynamic reaction force per mass, it is difficult to obtain high momentum from the water surface. The effect of the body density on the take-off velocity is also investigated. The take-off velocity, represented by black lines in the graph in Fig. 1c, varies with the Weber number at the same body density. In general, lower body densities are shown to be correlated with high jumping speed according to the analytical model, and the model results show a non-linear relationship between the dynamical scale and take-off velocity in the intermediate region between the surface tension-dominant region (We « 1) and drag-dominant region (We » 1). This observation suggests that the Weber number of the system should be much lower or much higher than 1 for effective jumping on water. Microscale jumping mechanisms[9,27,28] can be applied to jumping on the water in the surface tension-dominant region, but the jumping performance of robots in the drag-dominant region is improved more rapidly than in the surface-tension dominant region as the Weber number increases. Therefore, jumping systems should be designed on a large scale with low body density to achieve high jumping performance on water surfaces. In terms of efficiency (jumping kinetic energy/stored input energy) in the drag dominant region, the breaking of the water surface causes high levels of the splash, leading to inefficient jumps due to energy dissipation. On the other hand, a surface tension-based robot design is more suitable for high efficiency because the surface tension-based jump (We < 1) minimizes the splash on the water[9].

The analytical model and biological data indicate that water-jumping robots should be designed with a minimum system density

and a Weber number far from 1, surface tension dominant (We « 1) or drag dominant (We » 1), for high performance and load capacity. The Weber number represents not only the morphological scales associated with the size of the system but also the dynamic scale associated with the speed of the legs. Springtails and pygmy mole crickets that employ catapult mechanisms[29,30] have a higher Weber number than their morphological scale due to their faster leg speed (Fig. 2a). For example, pygmy mole crickets are smaller than water striders and fisher spiders but jump higher on water, exhibiting a high Weber number (We > 100). Pygmy mole crickets have higher leg tip velocities ($V_{leg}$ = 6.12 m/s) than fisher spiders ($V_{leg}$ = 0.4 m/s) or water striders ($V_{leg}$ = 0.15 m/s) using a catapult mechanism which is one of the power amplification mechanisms. The higher leg speed of these insects when they jump maximizes the Weber number, resulting in high jumping performance through drag-based propulsion. Insects with a high Weber number tend to manifest a higher surface area at the end of their legs. Pygmy mole crickets expand paddles and spurs attached to the end of their legs, and basilisk lizards spread out their feet to create a sort of paddle when jumping on water. They maximize the drag forces by expanding the surface area in contact with the water to take advantage of the relatively high leg speeds.

## Design of the water jumping robot
Based on the biological data and model analysis, we design a large-scale jumping robot (body length = 280–320 mm) with low body density and employ a catapult mechanism to increase Weber number, as shown in Fig. 2b. The robot is around 10 times larger than water striders (body length = 28 mm). A composite origami method[31–33] is employed for constructing a catapult mechanism[34] with a lightweight robot body (2 g). A shape memory alloy (>10 kW/kg) coil spring actuator, which is one of the actuators with the highest power density[35–37], is employed as the actuator of the robot. The pads at the end of the legs, called drag pads for generating drag, represent a significant morphological difference from the feet of insects with a low Weber number. The pads on the feet are fabricated in a semi-cylindrical shape with a glass fiber-reinforced plastic and a PET film; the feet have a large surface contact area but are lightweight to generate large drag force (Fig. 2c). The drag pads are coated with hydrophobic materials to reduce the pulling force that resists leaping out of the water surface in the jumping process[38]. A small rectangular sheet called the "lateral drag pad" is attached to the middle of the foot to generate lateral drag force to perform directional jumping. A more detailed description of the fabrication process of the robot is provided in Supplementary Fig. 1.

## Experimental validation of the theoretical modeling
We built three prototype robots that employed the same actuation mechanism but were designed with different foot shapes and sizes to jump with different Weber numbers. The specifications of prototypes are listed in Supplementary Table 1. The experiments show that jumping performance on the water depends on the Weber number. From high-speed videos, robot A rapidly moves its feet downwards, making air pockets, and then the feet come out of the water before the air pockets close (Fig. 3a, Supplementary Movie 1). The robot jumps up to 545 mm on the water surface (Fig. 3b). The hydrodynamic forces exerted on the foot while the robot takes off are calculated by the position data of the foot in the vertical direction (Fig. 3c). The vertical drag force is the largest propulsive force during take-off and is approximately 131 times larger than the surface tension (We = 131). The take-off velocity of robot A is measured at approximately 3.6 m/s based on the time zero intercept of the trend of the vertical velocity over time (Fig. 3d). This vertical jumping performance is superior to that of existing water-jumping animals and previous robots (Fig. 1c). For comparison, robot B with punched drag pads (Fig. 3e), which has the same body and power density but a smaller vertical projected area of feet, exhibits worse performance. Its Weber number and take-off

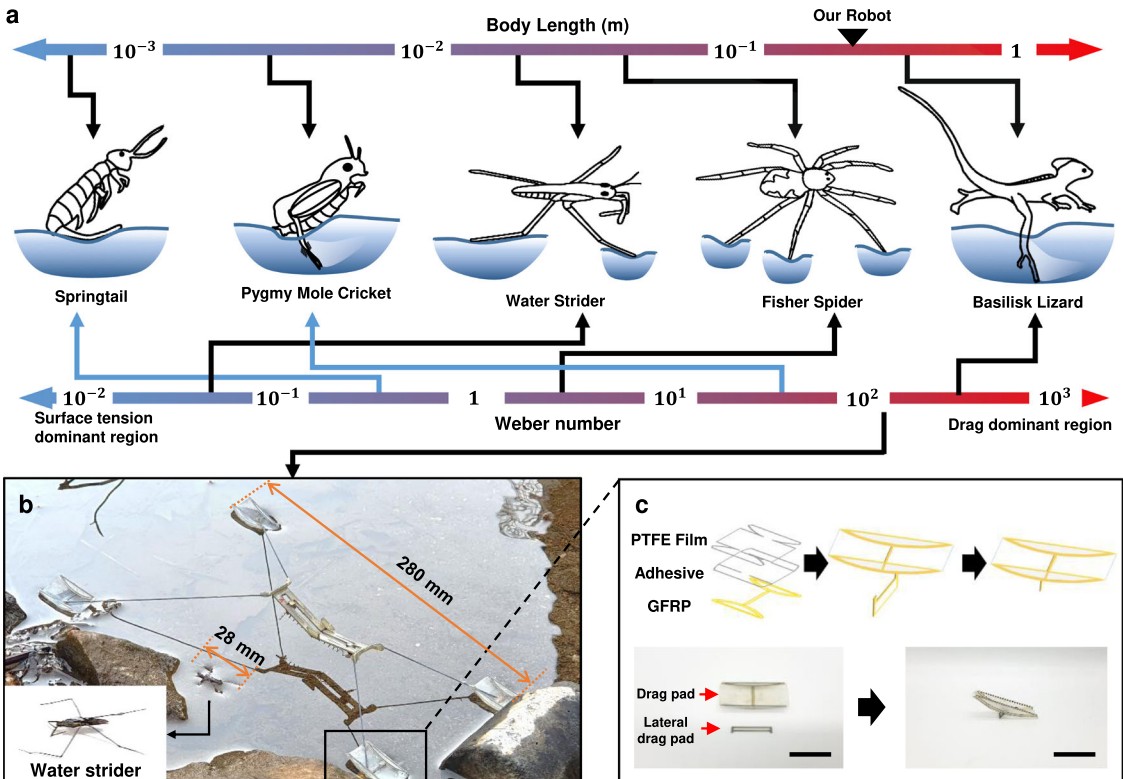

**Fig. 2 | Scale of the semi-aquatic organisms and morphological design of the robotic insect. a** A relationship between the body length and Weber number of animals jumping on water. **b** A large-scale robotic insect consisting of a shape memory alloy spring actuator and four semi-cylinder feet. The robot is designed large size (Body length = 280 mm) that is ten times larger than a water strider (Body length = 28 mm) for drag propulsion. **c** Lamination and folding steps of the drag pad. A lateral drag pad for the directional jump is attached to the bottom of the forefoot. Scale bar, 20 mm.

velocity are measured at 34 and 2.4 m/s, respectively (Supplementary Fig. 2a, b). Robot C has wire-type legs and is designed half the size of robots A and robot B to jump with a low Weber number in the intermediate-scale region. Its Weber number is about 6, and the take-off velocity is 1 m/s (Supplementary Fig. 2c, d), which is the lowest performance among the robot prototypes. The take-off velocity decreases with increasing momentum obtained from surface tension than drag force (Fig. 3d–f). It indicates that the jumping performance varies depending on the morphological shape that determines the Weber number. As shown in the simple model analysis, results show that the robot with a high Weber number tends to jump higher on the water in the drag-dominant scale region(Supplementary Movie 2), and the robot with a Weber number close to 1 is not able to jump on the water even though it has high actuation power.

## Asymmetric design for long and high jumping on water

To take advantage of jumping locomotion that can overcome obstacles larger than the jumper itself, a design for directional jumping as well as vertical jumping is needed. Another advantage of drag-based propulsion is that the lateral drag force can be easily adjusted by changing the size of the lateral drag pad, which changes the projected area, and by changing the leg length, which changes the speed of the driving leg. Two robot prototypes with different design parameters, namely, a symmetrically designed jumping robot (SJR) and an asymmetrically designed jumping robot (AJR), are tested on the water surface to determine which exhibits more efficient directional jumping (Supplementary Table 2). The length of all the legs of the SJR is 130 mm, and the lengths of the front legs and hind legs of the AJR are 130 mm and 65 mm, respectively (Supplementary Fig. 3a, b). According to the experimental data, the AJR has approximately twice the angular velocity of the front legs compared to the hind legs

(Supplementary Fig. 3c–f). Figure 4 shows the experimental results of SJRs and AJRs with different lateral drag pad heights. Figure 4a, b shows the position, vertical velocity, and lateral velocity of the SJR and AJR. The data are obtained by tracking the center of the body of the robots (Fig. 4c). After take-off, the vertical velocity decreases because of gravity, and the lateral velocity decreases due to aerodynamic drag. The initial jumping angle should be designed close to 45° to achieve the longest jump. However, the take-off velocity of the robot gradually decreases as the area of the lateral drag pad increases due to the energy dissipation to water flow (Fig. 4e, f). Thus, the trade-off between take-off velocity and initial jumping angle must be optimized. A comparison of the SJR with 2.6 mm lateral drag pads and the AJR with 2.1 mm lateral drag pads, which have the same take-off velocity, shows that the AJR has an initial jumping angle closer to 45°. This means that the AJR is designed to leap forward at a more appropriate angle with the same energy dissipation. Therefore, the longest jump distance is obtained with the optimal lateral drag pad size and robot design considering the take-off speed and angle (Fig. 4g). The AJR with 2.6 mm lateral drag pads jumps forward 556 mm in the lateral direction. The AJR robot can overcome 220 mm high obstacles, as shown in Fig. 4d (Supplementary Movie 1). The jumping angle of robots with drag-dominant propulsion can be adjusted with additional pads attached to their front legs, and robots jump farther with less energy consumption by rotating their front legs faster than their hind legs.

## Discussion

In this paper, we demonstrate robotic insects jumping in vertical and horizontal directions with high jumping performance through drag-based propulsion. The maximum performance of a system jumping from the water surface is determined by the hydrodynamic reaction force associated with the scale of the system, as demonstrated through

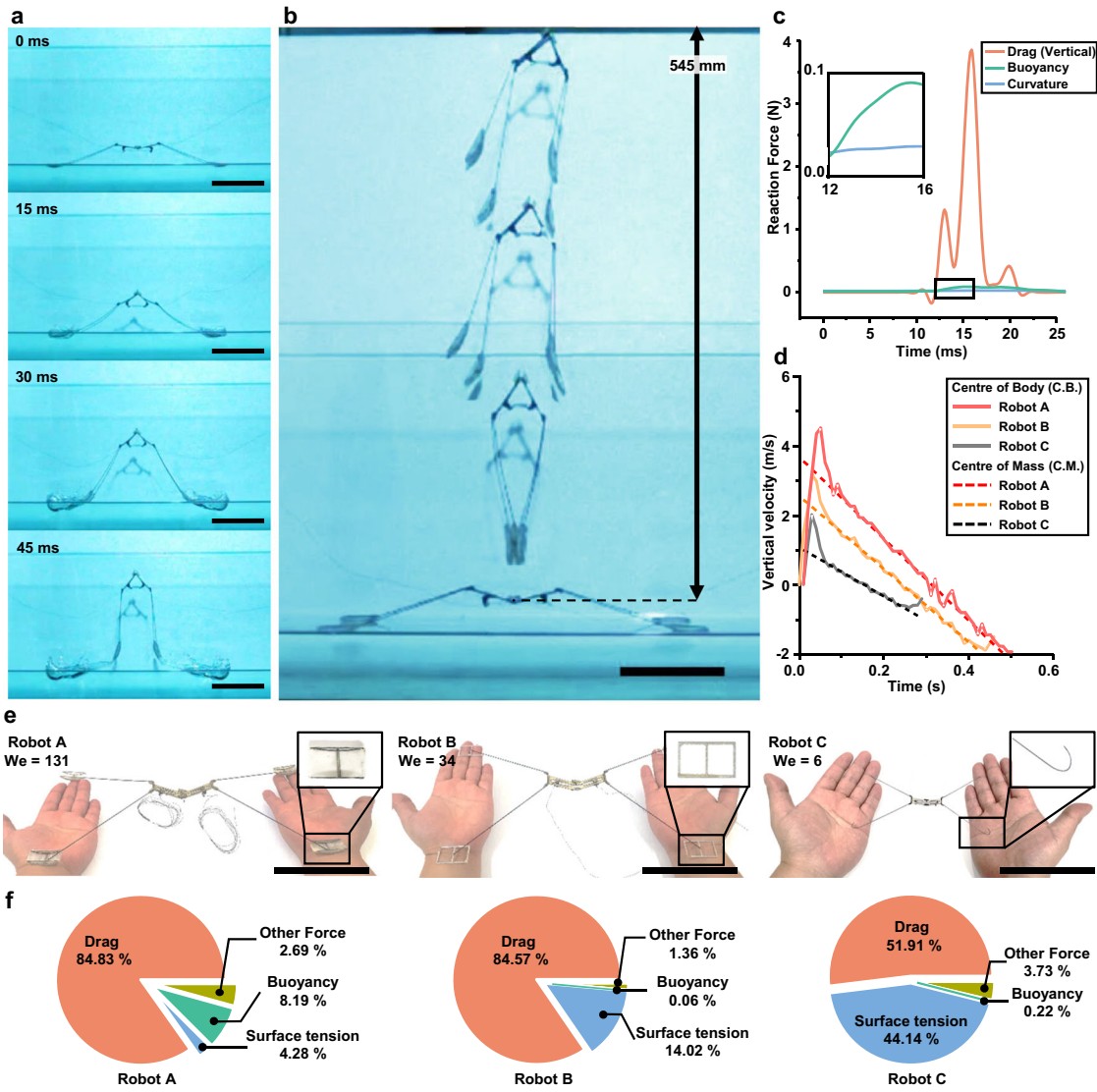

**Fig. 3 | Experimental results of vertical jumping on the water surface. a** Jumping sequence of the vertical jumping robot (Robot A). Scale bar, 10 cm. **b** Composite image of robot A jumping on the water surface. The robot jumps at 545 mm on the water's surface. Scale bar, 10 cm. **c** The change of drag, surface tension, and buoyancy force while robot A takes off on the water. Only the vertical force components are considered. **d** The vertical velocity of robot A, robot B, and robot C during the jump. Solid lines indicate the center of the body (C.B.), and dash lines indicate the velocity of the center of mass (C.M.) estimated from the oscillation curves of the center of body. **e** Images of a vertical jumping robot with semi-cylinder drag pads (Robot A), a robot with punched rectangular drag pads (Robot B), and a robot with wire-type feet (Robot C). Scale bar, 10 cm. **f** The momentum ratio of robot A, robot B, and robot C obtained from the water.

a simplified jumping model. Systems in the intermediate-scale dynamics region (We ~1) defined by the Weber number cannot achieve larger momentum than those in the drag-dominant region and surface tension-dominant region because of the limitation of the hydrodynamic reaction forces. Surface-tension-dominated propulsion is useful for small-scale jumping robots, but their limited jumping performance and payload limit practical engineering applications. A water-jumping robotic insect developed in this study can jump vertically on the water at a take-off velocity of 3.6 m/s. Adjustment of the leg length and the vertically projected area of the robot enables a jump of 556 mm in the horizontal direction and a jump over a 220 mm-high obstacle on the water surface.

The directional jumping experiments show that the robot is able to jump over the obstacles by generating vertical and lateral momentum from the drag pad, but the study has limitations in consecutive jumping and controllability of the jumping angle. To control the jumping direction of the robot, the initial body posture also can be an important parameter[39]. The additional component for adjusting the initial posture of the insect-scale robot by moving the center of mass or asymmetric hydrodynamic force may change the jumping angle of the robot. Another challenging issue for repeatable jumping is aerial righting and landing control of the insect-scale robot at the air-water interface. The body shape and mass distribution of the robot can control the orientation of the system in mid-air for landing by facing its ventral side to the water surface[39]. Future work attempts to realize consecutive jumping on the water surface by employing the landing mechanism of springtails and utilizing an additional antagonistic actuator capable of restoring the robot to its initial state.

A simplified model and simulations in this study verify that the scale-dependent hydrodynamic force determines the maximum jumping performance of various water-jumping robots and natural organisms. The design of water-jumping robots should be optimized by considering morphological and dynamical scales, and the maximum jumping height at a specific body density can be estimated. The experimental results show that the robot can jump over an obstacle through high jumping performance on a water surface and

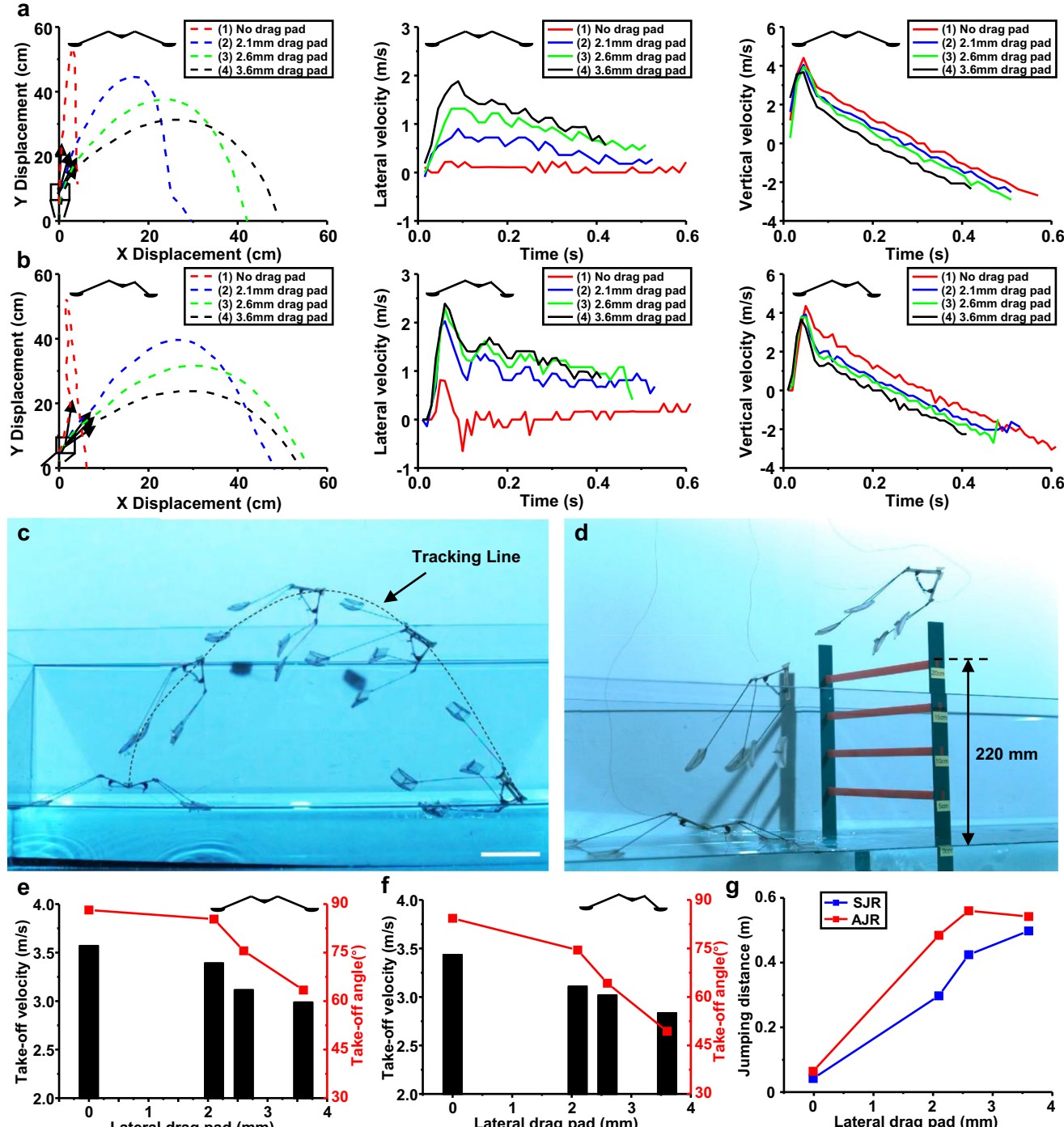

**Fig. 4 | Experimental results of lateral jumping on the water surface.**
**a**, **b** Jumping trajectory, lateral velocity, and vertical velocity of SJR (**a**) and AJR (**b**). The robots are tested by varying the height of lateral drag pads with the same width of 20 mm. **c** Jumping trajectory of the AJR, which attached 2.6 mm drag pad. Scale bar, 10 cm. **d** The AJR overcomes an obstacle of up to 220 mm height. **e**, **f** Take-off angle and take-off velocity of SJR (**e**) and AJR (**f**) depending on the height of lateral drag pads. **g** Lateral jumping distance depending on the height of lateral drag pads.

improve our understanding of the dynamic interaction between an unconstrained system and a water surface during jumping. The results also provide insight into understanding the evolution of organisms that have a high jumping capability on water.

## Methods
### Model Analysis
We examine the jumping performance depending on the scales using a simplified quasi-static model consisting of three components: leg ($L$),

foot ($L^2$), and body with actuators ($L^3$). In this model, the length of the legs, the area of the feet, and the volume of the body increase constantly with body length ($L$). These model components allow us to calculate the hydrodynamic reaction forces in terms of the scales of the system.

The leg is represented by springs and is massless. The acceleration time to reach the take-off velocity from rest is limited by the length of the leg[8]. Assuming that the mean speed of the body during take-off is $v/2$ under constant acceleration conditions[40], the time available for

extending their legs is $2L/v$. From the impulse-momentum theorem, the take-off velocity of the body can be expressed as

$$v_{\text{body}} = \sqrt{\frac{2FL}{m}} \tag{1}$$

The jumping animals and robots rely on the hydrodynamic reaction force acting on their legs or feet. Based on the obtained data in Fig. 1, jumping locomotion animals and robots are dominated by surface tension force or drag force rather than buoyancy force (Bo < We). Buoyancy is ignored because the force is small in insect scales, and increasing the volume of the feet causes performance degradation due to the increased mass. The viscous force is also negligible compared with the hydrodynamic pressure force because most water walker is characterized by a high Reynolds number[20]. Considering dominant hydrodynamic forces according to their scales, the hydrodynamic reaction force ($F_{\text{hy}}$) can be simply expressed as the summation of inertia drag ($F_d$) and surface tension force ($F_s$). The hydrodynamic reaction force is written as

$$F_{\text{hy}} = F_d + F_s = 4\sigma L + 0.5 C_d \rho_w L^2 \left(v_{\text{leg}}\right)^2 \tag{2}$$

Where the $\sigma, C_d, \rho_w$ is the surface tension of water, foot-drag coefficient, and water density, respectively. The speed of the leg ($v_{\text{leg}}$) is assumed to be the speed of the muscles or actuators that contributed to the work of the jump. If the power density of the muscle or actuator is independent of body length, velocity of the leg can be expressed as $v_{\text{leg}} = \varepsilon L$ ($v_{\text{leg}}$ is velocity of leg, $\varepsilon$ the strain rate of muscle) because the muscle force is typically proportional to cross-section area ($F_{\text{muscle}} \sim L^2$)[40]. Then the reaction force considering the body mass can be written as

$$F = 4\sigma L + 0.5 C_d \rho_w \varepsilon^2 L^4 - mg \tag{3}$$

Therefore, Weber number, maximum output power ($P_{\text{max}}$), and take-off velocity ($v_{\text{body}}$) can be written as

$$We = \frac{F_d}{F_s} = \frac{C_d \rho_w \varepsilon^2 L^3}{8\sigma} \tag{4}$$

$$P_{\text{max}} = F_{\text{hy}} v_{\text{leg}} = 4\sigma \varepsilon L + 0.5 C_d \rho_w \varepsilon^3 L^5 \tag{5}$$

$$v_{\text{body}} = \sqrt{\frac{8\sigma L^2 + C_d \rho_w \varepsilon^2 L^5 - 2mgL}{m}} \tag{6}$$

The mass of the body can be expressed as $m = \rho_b L^3$ ($\rho_b$ is body density). Therefore, the take-off velocity can be written in terms of $L$ as

$$v_{\text{body}} = \sqrt{\frac{1}{\rho_b}\left(\frac{8\sigma}{L} + C_d \rho_w \varepsilon^2 L^2\right) - 2gL} \tag{7}$$

The Weber number is proportional to the volume of the systems ($L^3$) and proportional to the square of strain rate ($\varepsilon^2$). It indicates that the Weber number can be represented by the morphological scales and dynamical scales of the water jumping systems. The take-off velocity of the systems is determined by the body density, strain rate, and body length. If the body density and the strain rate is constant, take-off velocity increases as the system gets smaller or bigger because the velocity scales as $L^{-1}$ at a small scale and scales as $L^2$ at a large scale. Therefore, the system designed in an intermediate dynamical scale

region (We -1) has lower jumping performance than other scale regions.

Our assumption is that the muscle or actuator force can exert as much as the maximum hydrodynamic forces. The hydrodynamic reaction force scales as $L^4$, requiring that the muscle force is proportional to $L^4$ to exert the maximum hydrodynamic forces. However, the muscle typically scales as $L^2$; muscles or actuators cannot exert a larger force than hydrodynamic reaction force in a large-scale region ($L \gg 1$). In the region where the hydrodynamic reaction force is larger than the muscle force, the jumping velocity does not increase because the reaction force is limited by muscle force. However, in insect scales where the muscle force can exert larger than the hydrodynamic force, the system dynamics are dominated by hydrodynamic reaction forces. Therefore, the take-off velocity varies with the size of systems from the take-off velocity equation above. Based on the take-off velocity equation, jumping height ($H$) can be calculated as the following equation.

$$H = \frac{v_{\text{body}}^2}{2g} = \frac{1}{2g\rho_b}\left(\frac{8\sigma}{L} + C_d \rho_w \varepsilon^2 L^2\right) - L \tag{8}$$

In addition, size-normalized jumping height also clarifies the scale-dependency of the water-jumping performance by the dominant hydrodynamic force that the system can get in different scales. The jumping height normalized by body length ($L$) can be expressed as

$$\frac{H}{L} = \frac{1}{2g\rho_b}\left(\frac{8\sigma}{L^2} + C_d \rho_w \varepsilon^2 L\right) - 1 \tag{9}$$

The jumping height normalized by body length has a non-linear relationship with the Weber number (Supplementary Fig. 4). Unlike the jumping height graph in Fig. 1c, the slope is steeper in the surface tension-dominant region than in the drag-dominant region. The robot design for the low Weber number can be considered if the optimization objective of the jumping is efficiency and normalized height improvement, rather than absolute jumping height improvement.

## Fabrication of the main body

The composite membrane origami method is a fabrication process to build a multi-material layered structure that integrates a rigid link made of high-rigidity materials with flexible joints[31–33]. The process allows for the fabrication of lightweight, high-rigidity structures with appropriate material selection. The main body consists of the rigid body, polymer layer, and adhesive layer. It is manufactured by laser cutting and laminating process. For lightweight, glass fiber reinforced plastics (GFRP) were used as the rigid body, and PET film was used as the polymer layer, as shown in Supplementary Fig. 1a. The spring-loaded joint parts are replaced by high-stiffness aramid flexures. Aramid Flexure is made of integration of GFRP, TPU, and aramid fibers. To utilize TPU as an adhesive layer, the composite is fabricated by a heating press machine at 200 degrees Celsius. Shape memory alloy (SMA) spring is employed as the extensor actuator considering its high energy density and capability of large deformation. This fabrication process and design make it possible to release impulse force with a simple structure.

All robots except robot C used the same actuator and main body. The coil and core diameter of the SMA Spring used as an actuator are 0.381 mm and 1.5 mm, respectively. The number of turns is 20, and the initial length after the training of the SMA spring is 12 mm. The main body of the robot has a size of 21 mm × 79 mm and weighs 2 g. For robot C, the main body and SMA spring were designed at half scale to reduce the Weber number. The coil diameter, core diameter, and initial

length of the SMA spring for robot C is 0.203 mm, 0.8 mm, and 6 mm, respectively. The main body of robot C is 10.5 mm × 30.5 mm and weighs 0.4 g.

## Fabrication of the leg and feet

The legs of the robots should also be made light, and the feet should be designed to have a high lifting force. For high rigidity and lightweight, the legs are made of carbon rod with a diameter of 0.8 mm. The feet of robots A, SJR, and AJR are designed in a semi-cylinder shape. The curved shape has a linear force profile depending on sinking depth[9]. The foot was made in a three-dimensional structure using the origami method (Fig. 2c). The foot frame is made of GFRP for light weight with high stiffness, and PTFE is used for the film that met the water surface. Lateral drag pads are attached to the front feet of the robot. The final assembled robot is shown in Supplementary Fig. 1b. For comparison of the jumping performance depending on Weber number, the drag pad of robot B was designed as a punched square pad made of GFRP (Supplementary Fig. 2a, b), and the feet of robot C were made of 0.305 mm diameter curved super-elastic SMA wire (Supplementary Fig. 2c, d). The super-hydrophobic coating is applied to the surface of the feet since super-hydrophobicity increases upward force and reduces energy consumption in the process of jumping on water[38].

## Measurements of and experimental on the robot

Robot jumping experiments were conducted in a 90 cm × 30 cm × 30 cm aquarium. A shape memory alloy spring actuator is connected by external wires and heated by a power supply of 0.8 A. The jumping videos were recorded by two high-speed digital cameras (Phantom MIRO EX4 and MIRO C320, Vision Research, USA) at 1000 frames per second. Position data were obtained by the vision analysis program (Kinovea). Two tracking points were put on the center of the body and the center of the front foot. The body and pad velocities were computed from data collected from tracking points, and the virtual center of mass of robots was measured based on the trend of the average oscillation curves obtained by the tracking point of the center of the body.

## Data availability

Data supporting the findings of this study are available within the paper. All other relevant data are available from authors upon request. Source data are provided with this paper.

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

## Acknowledgements
This work was supported by the Ajou University research fund.

## Author contributions
M.G., S.H., D.Kang, and J.-S.K. initiated the project and discussed the results. M.G., D.Kim, and B.K. designed and conducted the research. S.H., D.Kang, and J.-S.K. supervised the work, provided feedback, and prepared the paper. All authors revised the paper.

## Competing interests
The authors declare no competing interests.
