## [Peer Review File · Nature Communications]

REVIEWER COMMENTS

Reviewer #1 (Remarks to the Author):

So, I have to begin this review with a statement of bias on my part. I am a specialist in the field and this paper, from my opinion, is of great interest to specialists in the field. Overall, the interface of the biological and robotic data combined with the application of the relevant dimensionless parameters look to me to be of great interest to specialists.

My extreme positive bias toward this paper as a specialist makes it difficult for me to determine if this paper would be of interest to a 'general audience'. I must admit an agnostic stance on this, I do apologize. I do think this paper is of extreme value for specialists and support its publication highly.

In particular, the application of the dimensionless analysis to jumps from water (particularly the Weber number discussion) is not something I've seen done before to this detail. If this manuscript goes to press, figure 1C will be going into my lectures.

I have only small points on this paper.

1) Figure 1, please put slopes on the lines in figures 1A and B. The data is in the legend, but the slopes are so striking it may be better to have them on the figure.

2) Figure 2: to my knowledge the pygmy mole cricket and the basilisk lizard both break the surface of the water during their motions. Please reflect this in the drawings up top (it looks like they don't break the surface in the drawings in the current manuscript)

3) Extended Table 1: Please group the jumping vs the running animals. It's clear from the paper and the text what you're talking about, but the table, as shown, has a (running) just hanging in the first column of the figure. A slight re-build of the table would make what you're talking about much clearer.

4) Extended table 1: there's some strange things happening with significant figures, the basilisk lizard Weber number is certainly not known to 5 significant figures. Please adjust your significant figures to be appropriate.

5) Extended data figure 2: I think this data is important enough to be included in the main manuscript (with the small caveat that I think the red/green color scheme for drag vs curvature force - I think this is surface tension generated? - should be changed for red/green colour blind individuals).

These 5 points are all extremely small points that all stem from my extreme interest in this manuscript, I leave the authors to consider which they modify. The only mandatory one is number 4, which should be trivial to change for the authors.

Reviewer #2 (Remarks to the Author):

In this paper, the authors investigate the scale dependent momentum transfer of jumping mechanisms. They find that having a larger Weber number leads to larger jumping height and distance. Based on their models, they design and build a jumping robot and demonstrate a maximum jumping height of 545 mm and a distance of 556 mm. These exceed prior jumping robots. Overall, I find the analysis to be interesting, thorough, and convincing. The robotic demonstration is also well presented. In my opinion, the paper is suitable for publication in Nature Communications if the authors can address my major and minor comments below.

Major comments:

1. Throughout the paper, the authors aim to analyze jumping performance based on 3 non-dimensional numbers: Ba , We , and Bo . The analysis is easy to follow and well described. However, I feel the optimization objective (jumping height and distance) may need reconsideration or at least justification. It feels intuitive to the reviewer that larger, drag-based jumpers will reach a larger jump height because it can store more energy and have a larger contact area with water. Should the optimization objective also be some non-dimensionalized number, such as jumping height normalized by body length? In my opinion, this may give a fair comparison in contrast to using the absolute distance, which seems to penalize smaller robots/animals.
2. As shown in Extended Data Table 1, the robot jump height exceeds that of animals, which is impressive. However, can the authors also compare the jumping energetics? How efficient is the robot jumping (Energy in J/m^2g^*h)? When insects jump in nature, are they trying to optimize jump height or efficiency or some weighted average of several metrics?
3. While the robot jumps very high, there is no feedback control in the jumping process. Can the authors discuss whether implementing control would be helpful for jumping? Following up on this point, the authors demonstrate directional jumping. In that case, the robot is fabricated with asymmetric legs so it

jumps in one direction. This seems limiting. Can the authors comment on how future design can incorporate either new mechanisms or control to enable adjustable directional jumping? I do not require new experiments to be done, but I think adding a discussion can strengthen the paper.

4. Another issue with the current robot design is lacking the ability to land on the water surface and jump again. I think this is important from a robotic application perspective. I understand the paper's goal is to understand the physics of jumping (on the water surface) and this may not be a focus of the study. I feel a discussion on controlled landing and consecutive jumping can further strengthen the paper.

Minor comments:

- I can't find where Extended Data Figures 1-2 are cited
- There are minor formatting issues in several places. For instance, there lacks a space between 545 and mm in line 32. Similar issues exist in lines 445, 465 and Extended Data Fig. 3 a-b
- In line 414, please change "get" to "gets"

Response to Reviewer #1 Comments

So, I have to begin this review with a statement of bias on my part. I am a specialist in the field and this paper, from my opinion, is of great interest to specialists in the field. Overall, the interface of the biological and robotic data combined with the application of the relevant dimensionless parameters look to me to be of great interest to specialists. My extreme positive bias toward this paper as a specialist makes it difficult for me to determine if this paper would be of interest to a 'general audience'. I must admit an agnostic stance on this, I do apologize. I do think this paper is of extreme value for specialists and support its publication highly. In particular, the application of the dimensionless analysis to jumps from water (particularly the Weber number discussion) is not something I've seen done before to this detail. If this manuscript goes to press, figure 1C will be going into my lectures.

Response: We appreciate the reviewer's valuable comments for identifying significance of this research. We hope that the new perspectives of jumping on water related to dimensionless parameters at the air-water interface would be valuable and interesting issue for specialists in this field. Our analysis of the scale-dependent hydrodynamics of water jumpers will be a useful framework for understanding nature and robotic systems interacting with water surface. We sincerely appreciate your kind comments and have carefully revised our manuscript according to your feedback.

Comment 1: Figure 1, please put slopes on the lines in figures 1A and B. The data is in the legend, but the slopes are so striking it may be better to have them on the figure.

Response: Thank you for your suggestion. We agree that the slope on the line will help readers better understand the trend of nondimensional numbers in Fig. 1. We revised Fig. 1a-b.

Revised Figure in manuscript:

(Fig. 1.)

Fig. 1. The trend of the Baudoin (Ba), Bond (Bo), and take-off velocity of jumping animals and robots depending on Weber (We) number. a, b Baudoin (Ba), Weber (We), and Bond (Bo) number for insects and robots jumping on water. Bold lines indicate the dependence of the best fit line ($Ba = 0.0496 We^{0.54}, r^2 = 0.597, Bo = 0.0038 We^{0.99}, r^2 = 0.7221$) and dashed line indicate the $Ba=We$ and $Bo=We$, respectively. Blue region indicates that the jumping locomotion of systems are dominated by surface tension rather than drag ($We \ll 1$), the systems locomotion in red region are dominated by drag rather than surface tension. c The take-off velocity of the insects

and robots depending on Weber number. Colour contour indicates the body density(kg/m^3) on the Weber number simulated by a simple model consisting of three components: body(L^3), foot(L^2), and leg(L). Black lines indicate that the take-off velocity of the model with various body density varies with Weber number. Data for characteristic leg speed U , area A , width w , and the take-off velocity collected from: (1) Yang et al. (2016), Hu et al. (2010) and Hu et al. (2003) (2) Suter et al. (2000) and Shin et al. (2008) (3) Sudo et al. (2015), Hu et al. (2010) and Kim et al. (2017), (4) Burrows et al. (2012), (5) Bush et al. (2006), Hsieh et al. (2003) and Glasheen et al. (1996) (6) Shin et al. (2008), (7) Yang et al.(2016), (8) Hu et al.(2010), (9) Koh et al. (2015).

Comment 2: Figure 2: to my knowledge the pygmy mole cricket and the basilisk lizard both break the surface of the water during their motions. Please reflect this in the drawings up top (it looks like they don't break the surface in the drawings in the current manuscript)

Response: This is a very thoughtful comment. We agree that the broken water surface should be reflected in the figure because the Weber number is strongly related to the state of the water surface during jump on water. As you mentioned, the driving legs of basilisk lizards²⁶ and pygmy mole cricket¹⁰ penetrate the water surface during their motions, and they are characterized by a high Weber number ($We > 1$). Hu et al. (J. Fluid Mech., 2010) found that the water surface is broken when the driving stroke is characterized by a $We > 1$. We think that the state of the water surface helps to understand the correlation between the Weber number and the scales of robots and organisms. We have redrawn Fig. 2 to represent the broken water surface.

Revised figure in manuscript:

(Fig. 2)

Fig. 2. Scale of the semi-aquatic organisms and Morphological design of the robotic insect. a A relationship between the body length and Weber number of animals jumping on water. **b** A

large-scale robotic insect consisting of a shape memory alloy spring actuator and four semi-cylinder feet. The robot is designed large size (Body length=280 mm) that is ten times larger than a water strider (Body length=28 mm) for drag propulsion. c Lamination and folding steps of the drag pad. A lateral drag pad for directional jump is attached to the bottom of the forefoot. Scale bar, 20 mm.

Comment 3: Please group the jumping vs the running animals. It's clear from the paper and the text what you're talking about, but the table, as shown, has a (running) just hanging in the first column of the figure. A slight re-build of the table would make what you're talking about much clearer.

Response: Thank you for your thoughtful comments. We have grouped the locomotion of animals and robots in Supplementary Table 1.

Revised table in Supplementary information:

(Supplementary Table 1.)

Species	Locomotion	L (mm)	M (mg)	V (m/s)	Ba	We	Bo	Re	Source
Insects									
Water Strider		28	4.59	1.1	0.03	0.06	0.0005	16	[1, 15, 24]
Fisher spider	Jumping	74*	233	0.6	0.092	1 ~ 10	0.01 ~ 0.1	100 ~ 1000	[3, 13]
Springtail		0.5	0.102	0.63	0.012	0.4	0.0014	50	[4, 15, 26]
Pygmy mole cricket		5.56	9.2	1.3	0.11*	57*	0.0017*	670*	[10]
Basilisk lizards	Running	87*	20800	1.4	22*	670*	20*	24000*	[20, 23, 26]
Robots									
Robot 1		25*	510	0.09	0.200	4.700	0.0054	260	[13]
Robot 2	Jumping	140	10200	1.92	1.4*	1000*	5*	66000*	[25]
Robot 3		13	4.08	0.5 ~ 1.0	0.013	7 ~ 28	0.6	1000	[15]
Robot 4		95*	68	1.67	0.060	0.165	0.001	48	[9]
This study									
Robot A		320	3000	3.6	1.021	131	1.225	3560	
Robot B	Jumping	320	3000	2.46	0.427	34.340	0.081	3560	
Robot C		160	680	1	0.075	6.050	0.0049	134.1	

Supplementary Table 1. Data sheet of the animals and robots jumping on water. L represents body length (mm) including the leg length, *M* represents mass (mg), *V* is jumping velocity (m/s), Ba is Baudoin number, We is Weber number, Bo is Bond number, and Re is Reynolds number. Values with * symbols in the blue are approximated values calculated from videos and images in the literature.

Comment 4: Extended table 1: there's some strange things happening with significant figures, the basilisk lizard Weber number is certainly not known to 5 significant figures. Please adjust your significant figures to be appropriate.

Response: We appreciate the reviewer's comment. The Weber number is a dimensionless characteristic value, so it is not necessary to have 5 significant value. All data for animals were corrected to have 2 significant figures in Supplementary Table 1 as shown revised table above.

Comment 5: Extended data figure 2: I think this data is important enough to be included in the main manuscript (with the small caveat that I think the red/green color scheme for drag vs curvature force - I think this is surface tension generated? - should be changed for red/green colour blind individuals).

Response: We appreciate the reviewer's recommendation. Authors agree that the experimental data of three robot designs is very important to verify our model and simulation data. The three robots (robot 'A', robot 'B', and robot 'C') have the same actuation mechanism, but were designed with different foot shapes and sizes to jump with different Weber numbers. As in the result of the simple model analysis for predicting the maximum jumping performance, experiments show the robot with a high Weber number tends to jump higher on the water in the drag dominant scale region, and the robot with a Weber number close to 1 is not able to jump on water even though it has high actuation power. We combined the data of robot 'B' and robot 'C' contained in Supplementary figure 2 with Fig. 3, and moved the combined figure to the main manuscript. The colour in the figure is replaced by colour blind friendly palette (Tol_light) for red/green colour blind individuals. Thank you for the thoughtful and detailed suggestion.

Revised manuscript:

(page 8-9 line 162-186) We built three prototype robots that employed the same actuation mechanism but were designed with different foot shapes and sizes to jump with different Weber numbers. The specifications of prototypes are listed in Supplementary Table 1. The experiments show that jumping performance on water depending on the Weber number. From high-speed videos, robot 'A' rapidly moves its feet downwards, making air pockets, and then the feet come out of the water before the air pockets close (Fig. 3a, supplementary movie.1). The robot jumps up to 545 mm on the water surface (Fig. 3b). The hydrodynamic forces exerted on the foot while the robot takes off are calculated by the position data of the foot in the vertical direction (Fig. 3c). The vertical drag force is the largest propulsive force during take-off and is approximately 131 times larger than the surface tension ($We=131$). The take-off velocity of robot 'A' is measured at approximately 3.6 m/s based on the time zero intercept of the trend of the vertical velocity over time (Fig. 3d). This vertical jumping performance is superior to that of existing water-jumping animals and previous robots (Fig. 1c). For comparison, robot 'B' with punched drag pads (Fig. 3e), which has the same body and power density but a smaller vertical projected area of feet, exhibits worse performance. Its Weber number and take-off velocity are measured at 34 and 2.4 m/s, respectively. Robot 'C' has wire-type legs and is designed half the size of robots 'A' and 'B' to jump with low a Weber number in the intermediate-scale region. Its Weber number is about 6 and the take-off velocity is 1 m/s, which is the lowest performance among the robot prototypes. The take-off velocity decreases with increasing momentum obtained from surface tension than drag force (Fig. 3d, f). It indicates that the jumping performance varies depending on the morphological shape that determines the Weber number. As shown in the simple model analysis, results show that the robot with a high Weber number tends to jump higher on the water in the drag-dominant scale region, and the robot with a Weber number close to 1 is not able to jump on water even though it has high actuation power.

Revised figure in manuscript:

(Fig. 3)

Fig. 3. Experimental results of vertical jumping on the water surface. a Jumping sequence of the vertical jumping robot (Robot A). Scale bar, 10 cm. **b** Composite image of robot A jumping on the water surface. The robot jumps at 545 mm on the water surface. Scale bar, 10 cm. **c** The change of drag, surface tension, and buoyancy force while robot A takes off on water. Only the vertical force components are considered. **d** The vertical velocity of robot A, robot B and robot C

during the jump. Solid lines indicate the centre of body (C.B.), and dash lines indicate the velocity of the centre of mass (C.M.) estimated from the oscillation curves of the centre of body. **e** Images of a vertical jumping robot with semi-cylinder drag pads (Robot A), a robot with punched rectangular drag pads (Robot B), and a robot with wire-type feet (Robot C). Scale bar, 10 cm. **f** The momentum ratio of robot A, robot B and robot C obtained from the water.

Revised figure in Supplementary information:

Supplementary Fig. 3. Experimental results of vertical jumping robot ‘B’ and robot ‘C’. **a** Jumping sequence of robot B with punched rectangular drag pads. Scale bar, 10 cm. **b** Robot B jumps up to 313mm on the water surface. The Weber number of the robot is 34. Scale bar, 10 cm. **c** Jumping sequence of robot C with wire-type legs. **d** The robot C jumps up to 93 mm, and the Weber number of the robot is about 6. Scale bar, 10 cm.

Response to Reviewer #2 Comments

In this paper, the authors investigate the scale dependent momentum transfer of jumping mechanisms. They find that having a larger Weber number leads to larger jumping height and distance. Based on their models, they design and build a jumping robot and demonstrate a maximum jumping height of 545 mm and a distance of 556 mm. These exceed prior jumping robots. Overall, I find the analysis to be interesting, thorough, and convincing. The robotic demonstration is also well presented. In my opinion, the paper is suitable for publication in Nature Communications if the authors can address my major and minor comments below.

Response: Authors thank the reviewer for constructive comments. We revised the manuscript by addressing reviewer's thoughtful questions and comments. We hope our respond and revision would improve the quality of our paper for better understanding. Thanks for the detailed and thoughtful review.

Comment 1: Throughout the paper, the authors aim to analyze jumping performance based on 3 non-dimensional numbers: Ba , We , and Bo . The analysis is easy to follow and well described. However, I feel the optimization objective (jumping height and distance) may need reconsideration or at least justification. It feels intuitive to the reviewer that larger, drag-based jumpers will reach a larger jump height because it can store more energy and have a larger contact area with water. Should the optimization objective also be some non-dimensionalized number, such as jumping height normalized by body length? In my opinion, this may give a fair comparison in contrast to using the absolute distance, which seems to penalize smaller robots/animals.

Response: Thanks for the helpful comments. We agree that it is intuitive that a larger drag-based jumper will reach a higher jump height if the robot has the same power density. However, interestingly, the jump height decreases as the size of the robot increases in the specific scale region (surface tension dominant region, $We < 1$), even though it can store more energy, force, and velocity.

Therefore, we try to classify the dominant hydrodynamic force region that should be considered for designing a water jumping robot and understand the correlation between jumping performance and morphological shape of semi-aquatic animals.

The optimization objective in this research is to propose a design and prediction model to maximize the absolute jumping performance at the limited power density of the robots. In the revision process, we plotted the size-normalized jumping height as reviewer's suggestion, it shows much more dramatic nonlinearity which the paper aims to empathize with (See Supplementary Fig. 4). In addition, we wanted to include the size and the absolute jumping height in the modelling in order to present a limited momentum transfer depending on a dominant hydrodynamic force in each scale. Please understand that we can provide information about the limited momentum transfer with the absolute jumping height plot. We believe that these results provide a design guideline for roboticists designing water jumping robots to predict the absolute jumping performance depending on their dynamical and morphological scales.

We think the reviewer's perspective on normalized jump height and energy efficiency can provide readers with clear understanding as well. Normalized jumping height (Jumping height/body length) is much clearer to identify dominant hydrodynamic scale regions such as the surface tension-dominant, drag-dominant and the intermediate scale region. (See equations and a figure below). A surface tension-based robot design is more suitable if the optimization objective of the jumping is the normalized height improvement or high efficiency, because the slope in $We < 1$ region is steeper than $We > 1$ region and the surface tension-based jump ($We < 1$) minimizes the splash on the water⁹. We added the following equations and the figure for normalized jumping height in the method section and supplementary data. Thank you for the helpful suggestion.

Revised Methods in Manuscript:

(page 21-22 line 349-360) Based on the take-off velocity equation, jumping height (H) can be calculated as following equation.

$$H = \frac{v_{body}^2}{2g} = \frac{1}{2g\rho_b} \left(\frac{8\sigma}{L} + C_d\rho_w\varepsilon^2L^2 \right) - L$$

In addition, size-normalized jumping height also clarify the scale-dependency of the water-jumping performance by the dominant hydrodynamic force that the system can get in different scales. The jumping height normalized by body length (L) can be expressed as

$$\frac{H}{L} = \frac{1}{2g\rho_b} \left(\frac{8\sigma}{L^2} + C_d\rho_w\varepsilon^2L \right) - 1$$

The jumping height normalized by body length has a non-linear relationship with the Weber number (Supplementary Fig. 4). Unlike the jumping height graph in Fig. 1c, the slope is steeper in the surface tension-dominant region than in the drag-dominant region. The robot design for the low Weber number can be considered if the optimization objective of the jumping is efficiency and normalized height improvement, rather than absolute jumping height improvement.

Revised Supplementary information:

Supplementary Fig. 4. Normalized jumping height related to Weber (We) number. The graph shows a non-linear relationship between jumping height normalized by body length (L) and Weber number. The slope of normalized jumping height is steeper in the surface tension-dominant region than in the drag-dominant region.

Comment 2: As shown in Extended Data Table 1, the robot jump height exceeds that of animals, which is impressive. However, can the authors also compare the jumping energetics? How efficient is the robot jumping (Energy in / m*g*h)? When insects jump in nature, are they trying to optimize jump height or efficiency or some weighted average of several metrics?

Response: Thanks for the thoughtful comment on the important issue. First of all, jumping is very inefficient locomotion for animals and robots in terms of the cost of transport⁴⁰. Jumping is used for moving fast and escaping quickly from predators, though much more energy is consumed. The energy efficiency of animals and robots is determined by their jumping mechanisms and environments where they jump. We don't have energetic data of animals that jump on the water surface, but efficiency of the robot in this research can be calculated. The stored elastic energy of the SMA actuator is 0.135 J. Energy efficiency of the robot 'A' (jumping kinetic energy / stored input energy) is about 14.4% on the water surface. Authors suspect that animals have much lower efficiency than robots because animals have many parts not for jumping but for living.

In terms of energy efficiency depending on the dominant hydrodynamic forces, surface tension-based jumps are more efficient than drag-based jumps. The energy efficiency in $We < 1$ is similar to ground jumping because surface tension is a static force that can store the driving force like spring. In the drag-dominant region, the water surface is broken and leading to high levels of splash and flow that dissipate energy. However, the limited momentum transfer associated with the jumping performance should be considered first through the model that we proposed in Fig. 1. Robot 'A' and 'B' has the same power density and actuation mechanism, but the efficiency of robot 'B' ($V=2.4$ m/s, efficiency = 6.4%) is much lower than robot 'A' ($V=3.6$ m/s, efficiency = 14.4%) because the take-off velocity is limited by the maximum hydrodynamic reaction force acting on the foot, even if the muscles can exert sufficient force. The morphological size and shape represented by the Weber number highly affect the efficiency that jumping insects and robots that jump on water.

Unfortunately, we cannot find that the insects in nature try to optimize their dynamical and morphological scales or evolved for increasing efficiency. This is very interesting issue for further

study as future work. Thank you for the insightful question. The discussion on efficiency is added in the manuscript as follows.

Revised manuscript:

(page 6 line 125-129) In terms of efficiency (jumping kinetic energy / stored input energy) in the drag dominant region, the breaking of the water surface causes high levels of splash, leading to inefficient jumps due to energy dissipation. On the other hand, a surface tension-based robot design is more suitable for high efficiency because the surface tension-based jump ($We < 1$) minimizes the splash on the water⁹.

Comment 3: While the robot jumps very high, there is no feedback control in the jumping process. Can the authors discuss whether implementing control would be helpful for jumping? Following up on this point, the authors demonstrate directional jumping. In that case, the robot is fabricated with asymmetric legs so it jumps in one direction. This seems limiting. Can the authors comment on how future design can incorporate either new mechanisms or control to enable adjustable directional jumping? I do not require new experiments to be done, but I think adding a discussion can strengthen the paper.

Response: Thank you for your constructive comments on the controllability of the robot. In this paper, the purpose of this research is to determine the morphology of the robot and organisms for jumping on water depending on the scale of the system. The controllability of the robot is also very interesting issue on jumping locomotion. Directional jumping of the robot with asymmetric legs is an approach with limitations because the robot has only one jumping direction. To control the jumping direction of the robot, the initial body posture also can be important parameter as seen in the recent research³⁹. Inspired by this biological discovery on jumping behaviour, we can implement directional jumping by integrating an additional component on the robot that can adjust the initial body posture of the robot with a minimum degradation of the jumping performance. We anticipate that the shape memory alloy-based artificial muscle actuator [a] enables the

development of the additional component for adjusting the posture of the robot, which can realize the directional jumping of the robot.

Revised manuscript:

(page 11 line 229-240) The directional jumping experiments show that the robot is able to jump over the obstacles by generating vertical and lateral momentum from the drag pad, but the study has limitations in consecutive jumping and controllability of the jumping angle. To control the jumping direction of the robot, the initial body posture also can be an important parameter³⁹. The additional actuation for adjusting the initial posture of the insect-scale robot by moving the centre of mass or asymmetric hydrodynamic force may change the jumping angle of the robot. Another challenging issue for repeatable jumping is aerial righting and landing control of the insect-scale robot at the air-water interface. The body shape and mass distribution of the robot can control the orientation of the system in mid-air for landing by facing its ventral side to the water surface³⁹. Future work attempt to realize consecutive jumping on the water surface by adopting the landing mechanism of springtails and utilizing an additional antagonistic actuator capable of restoring the robot to its initial state.

Revised Reference:

39. Ortega-Jimenez, V. M., Challita, E. J., Kim, B., Ko, H., Gwon, M., Koh, J.-S. & Bhamla, M. S. Directional takeoff, aerial righting, and adhesion landing of semiaquatic springtails. *Proceedings of the National Academy of Sciences* **119**, e2211283119 (2022).

Reference for Response Letter

- [a] Kim, D., Kim, B., Shin, B., Shin, D., Lee, C.-K., Chung, J.-S., Seo, J., Kim, Y.-T., Sung, G., Seo, W., Kim, S., Hong, S., Hwang, S., Han, S., Kang, D., Lee, H.-S. & Koh, J.-S. Actuating compact wearable augmented reality devices by multifunctional artificial muscle. *Nature Communications* **13**, 1-13 (2022).

Comment 4: Another issue with the current robot design is lacking the ability to land on the water surface and jump again. I think this is important from a robotic application perspective. I understand the paper's goal is to understand the physics of jumping (on the water surface) and this may not be a focus of the study. I feel a discussion on controlled landing and consecutive jumping can further strengthen the paper.

Response: Thank you for your constructive comments about the landing of the robot. We agree that aerial stabilization and landing control, which enable repetitive jumping for practical applications of the robot, are very important. Though lots of studies are performed to improve the jumping performance of the robot, few studies have been conducted to develop robots capable of aerial righting and landing. Especially, limitations on the weight of the robot and actuator power restrict the usage of additional sensors or posture control units for aerial stabilization, while relatively large robots can do [b]. Therefore, aerial righting and landing control of the insect-scale robot still remains as a challenge. To address these issues, body shape and mass distribution of the robot can adjust orientation of the system in mid-air for near perfectly landing by facing its ventral side to the water surface, and it can be found in the recently published paper³⁹. In the future, we attempt to realize consecutive jumping on the water surface by adopting the landing mechanism of springtails and utilizing an additional antagonistic actuator capable of restoring the robot to its initial state.

Revised manuscript:

(page 11 line 229-240) The directional jumping experiments show that the robot is able to jump over the obstacles by generating vertical and lateral momentum from the drag pad, but the study has limitations in consecutive jumping and controllability of the jumping angle. To control the jumping direction of the robot, the initial body posture also can be an important parameter³⁹. The additional actuation for adjusting the initial posture of the insect-scale robot by moving the centre of mass or asymmetric hydrodynamic force may change the jumping angle of the robot. Another challenging issue for repeatable jumping is aerial righting and landing control of the insect-scale robot at the air-water interface. The body shape and mass distribution of the robot can control the

*orientation of the system in mid-air for landing by facing its ventral side to the water surface*³⁹.
Future work attempt to realize consecutive jumping on the water surface by adopting the landing mechanism of springtails and utilizing an additional antagonistic actuator capable of restoring the robot to its initial state.

Revised Reference:

39. Ortega-Jimenez, V. M., Challita, E. J., Kim, B., Ko, H., Gwon, M., Koh, J.-S. & Bhamla, M. S. Directional takeoff, aerial righting, and adhesion landing of semiaquatic springtails. *Proceedings of the National Academy of Sciences* **119**, e2211283119 (2022).

Reference for Response Letter

- [b] D. W. Haldane, J. K. Yim, and R. S. Fearing, “Repetitive extreme-acceleration (14-g) spatial jumping with Salto-1P,” 2017 IEEE/RSJ International Conference on Intelligent Robots and Systems (IROS). IEEE, Sep-2017.

Minor Comments:

- **I can't find where Extended Data Figures 1-2 are cited**

Response: Thank you for checking on this. The description of the Supplementary figures in our manuscript are as follows:

Revised manuscript:

*(Page 8, Line 160-161) A more detailed description of the fabrication process of the robot is provided in **Supplementary Fig. 1**.*

*(Page 8, Line 174-177) For comparison, robot 'B' with punched drag pads (Fig. 3e), which has the same body and power density but a smaller vertical projected area of feet, exhibits worse performance. Its Weber number and take-off velocity are measured at 34 and 2.4 m/s, respectively (**Supplementary Fig. 2a-b**).*

*(Page 8, Line 178-180) Its Weber number is about 6 and the take-off velocity is 1 m/s (**Supplementary Fig. 2c-d**), which is the lowest performance among the robot prototypes.*

*(Page 9, Line 194-197) The length of all the legs of the SJR is 130 mm, and the lengths of the front legs and hind legs of the AJR are 130 mm and 65 mm, respectively (**Supplementary Fig. 3a-b**). According to the experimental data, the AJR has approximately twice the angular velocity of the front legs compared to the hind legs (**Supplementary Fig. 3c-f**).*

*(Page 21, Line 356-357) The jumping height normalized by body length has a non-linear relationship with the Weber number (**Supplementary Fig. 4**).*

- There are minor formatting issues in several places. For instance, there lacks a space between 545 and mm in line 32. Similar issues exist in lines 445, 465 and Extended Data Fig. 3 a-b
- In line 414, please change “get” to “gets”

Response: We thank the reviewer for identifying this typographical error. All typos are corrected and revised carefully.

Revised manuscript:

(Page 15, Line 276-278) Fig. 3. Experimental results of vertical jumping on the water surface. a, Jumping sequence of the vertical jumping robot (Robot ‘A’). Scale bar, 10 cm. b, Composite image of robot ‘A’ jumping on the water surface. The robot jumps at 545 mm on the water surface. Scale bar, 10 cm.

(Page 17, Line 287-290) Fig. 4. Experimental results of lateral jumping on the water surface. a-b, Jumping trajectory, lateral velocity, and vertical velocity of SJR (a) and AJR (b). The robots are tested by varying the height of lateral drag pads with the same width of 20 mm. c, Jumping trajectory of the AJR which attached 2.6 mm drag pad. Scale bar, 10 cm.

(Page 20-21, Line 335-337) If the body density and the strain rate is constant, take-off velocity increases as the system gets smaller or bigger because the velocity scales as L^{-1} at small scale and scales as L^2 at large scale.

(Page 22, Line 377-378) The main body of the robot has a size of 21 mm x 79 mm and weighs 2 g.

(Page 23, Line 385-386) For high rigidity and lightweight, the legs are made of carbon rod with a diameter of 0.8 mm.

REVIEWERS' COMMENTS

Reviewer #1 (Remarks to the Author):

I thought the original manuscript was very strong, and still think so. The authors took all of my minor suggestions on board, and I'm very happy with the resulting manuscript.

Reviewer #2 (Remarks to the Author):

I thank the authors for their detailed response to my comments. I mostly agree with reviewer 1's comment on the paper quality and potential impact. In my view, the paper is of sufficient quality to be published in Nature Communications.

Response to Reviewer #1 Comments

Comment 1: I thought the original manuscript was very strong, and still think so. The authors took all of my minor suggestions on board, and I'm very happy with the resulting manuscript.

Our response: Authors thank the reviewer for the constructive comments and for recognizing the significance of this research. Your detailed suggestion has been invaluable in helping us to improve our paper, and we appreciate your time and effort in reviewing our paper.

Response to Reviewer #2 Comments

Comment 1: I thank the authors for their detailed response to my comments. I mostly agree with reviewer 1's comment on the paper quality and potential impact. In my view, the paper is of sufficient quality to be published in Nature Communications.

Our response: We thank the reviewer for valuable comments and suggestions. The quality of this paper has greatly improved thanks to the comments the reviewer provided. Thank you for your time and support in reviewing our paper.